# Investigating the Qualities of a Recreational Forest: Findings from the Cross-Sectional Hallerwald Case Study

**DOI:** 10.3390/ijerph17051676

**Published:** 2020-03-04

**Authors:** Renate Cervinka, Markus Schwab, Daniela Haluza

**Affiliations:** 1University College for Agrarian and Environmental Pedagogy, 1130 Vienna, Austria; renate.cervinka@haup.ac.at (R.C.); markus.schwab@haup.ac.at (M.S.); 2Department of Environmental Health, Center for Public Health, Medical University of Vienna, 1090 Vienna, Austria

**Keywords:** citizen science, green public health, green care, widen one’s mind scale, regional development, restorativeness

## Abstract

Prior research shows that forests contribute to human health and well-being. In this sense, this cross-sectional case study, adopting the principles of citizen science, assessed the restorative potential of places in the Hallerwald, an Austrian community forest. A convenience sample of adult forest visitors (n = 99, 64% females) completed a survey during a guided 2.5 h forest tour. The German questionnaire assessed the qualities of defined places in the forest. We also investigated changes in mood states, perceived stress, restoration, connectedness, and mindfulness before and after visiting the forest. In cooperation with a local working group, we developed the new Widen One’s Mind (WOM) scale, which showed good scale characteristics. All places received high scores in their potential to increase restoration and vitality and to widen one’s mind. Positive affect, restoration, connectedness with nature and the forest, and mindfulness increased pre- versus post-visits, whereas negative affect and perceived stress decreased. The findings of this study suggest that in recreational forests, visitors experience beneficial mental effects such as stress reduction in addition to physical exercise. To facilitate regional development goals, we recommend evaluating places in forests regarding the potential effects on the health and well-being as well as citizen participation before initiating extensive remodeling.

## 1. Introduction

The beneficial effects of visiting forests on human health and well-being are well documented [1,2,3]. Originating from East-Asia, the concept of Shinrin-Yoku, which means taking in the atmosphere of the forest, has reached Austria and received high attention in forestry and tourism in the late 2010s [2]. This concept inspired the local community of Adlwang in the Austrian province of Upper Austria to use a community forest, the Hallerwald, for future recreational purposes and tourism [4]. The initial idea was to envision the forest as a green space with dedicated places exhibiting different forest-specific characteristics. These places should be connected by a sign-posted path that allows for a comfortable walk that allows one to experience pleasure at the contact with nature all year-round. As an acknowledged recreational forest, the Hallerwald should provide recreation and vitality and inspire visitors to widen their mind. 

Following a co-creation approach, the local community invited scientific collaborators to evaluate the present situation in a case study, as so far, a base for regional development plans is lacking. In the Hallerwald case study, the local community enabled the further development of earlier empirical work of our research group on the value of forests for stress reduction, restoration, and health promotion [3,5]. The resulting collaboration reflects the principles of citizen science, a concept characterized by collaboration between scientists and members of the general public [6].

This study focused on a defined forest area related to Green Care, a current trend in Austria. Green Care aims at promoting diversification in agriculture and forestry as well as human health, well-being, and quality of life, which include educational, social, restorative, and therapeutic activities in nature [7,8]. These claims are also supported by the broader concept of Green Public Health, which is described as using all kinds of natural resources for public health promotion. These may encompass interventions on an individual or group level such as therapeutic horticulture and animal assisted interventions as well as on a community or population level [9]. Thus, the current study integrated several overlapping fields of action: regional development, Green Care, and Green Public Health.

A comprehensive body of literature shows that forests contribute to human health and well-being [1,3,10,11]. Forest visits provide restoration, decrease stress [12,13] and blood pressure [14], and improve reflection [15], mood [16,17,18,19,20], cognitive capacity [20], and vitality [21,22]. Restorative places in a forest increase connectedness with nature and the forest and mindfulness [5]. A Swedish study explored the non-timber values of recreational forests such as physical activity or gathering fruits of the forest (e.g., berries and mushrooms) [23]. Results showed that over 40% of the population would prefer to have a recreational forest in the vicinity of their homes. 

While there is abundant scientific evidence of the beneficial effects of forests, literature on the potential of forests to widen one’s mind is scarce. This concept refers to the idea of fostering contemplation, initiating change in perspective, and developing new perspectives and ideas from the forest visitors’ perception. Similar processes have been described in qualitative research on forest-based rehabilitation: patients’ mental processes changed from negative to positive thinking when choosing a favorite place in the forest, experiencing peace of mind and reflective thinking [24,25]. Established instruments for assessing reflective thinking have focused on measuring the characteristics of a person, not a place, hence insufficiently capture the notion of widening one’s mind. In a co-creation process, we developed a respective new scale with project partners from the local working group. 

Experiences in the forest are influenced by scene types [26], perceived sensory dimensions [13], and other characteristics such as the occurrence of water [27]. The current field study aimed at (I) assessing the qualities of typical places in the forest and their potential to increase recreation and vitality and to widen one’s mind, and (II) investigating changes in mood-states, perceived stress, restoration, connectedness, and mindfulness before and after visiting the forest (pre and post comparison). Based on prior research, we hypothesized that all places will reach high scores in their potential for increasing restoration and vitality and to widen one’s mind, and furthermore, that these scores will differ regarding the qualities of the various scenic characteristics [5]. Additionally, we hypothesized that positive affect, restoration, connectedness with nature and the forest, and mindfulness would increase during visits to the forest. In contrast, we expected a negative affect and perceived stress to decrease in the participants. To measure the place-specific potential for widening one’s mind, we developed the so-called Widen One’s Mind (WOM) scale and tested its psychometric properties.

## 2. Materials and Methods 

### 2.1. Design and Procedure

The study took place in the Hallerwald, a mixed forest with areas of monoculture (mainly spruces) covering a hilly terrain of 270,000 m^2^. The forest serves as a commercial forest, but regional development plans foresee an increase in the recreational use of the forest in the near future. Local partners from the community of Adlwang, where the Hallerwald is located, and researchers developed the study together, based on the concept of citizen science. We conducted the study in accordance with the principles laid down in the Declaration of Helsinki and the ethics guidelines of the state of Upper Austria. The project partners selected four distinct places that represented different scene types of the Hallerwald. Places should foster restoration and vitality and have the potential to widen one’s mind while being easily accessible by foot safely (e.g., without high risk of stumbling). 

Figure 1 shows the four places in the forest (Figure 1a–d): (A) Mossy stones embedded in a little creek (mossy stones); (B) Fern glade, a forest glade embossed by dense vegetation of ferns and horsetails enclosed by tall trees of a mixed forest (fern glade); (C) Outlook on the edge of the forest and bordered by a road, providing a view into a wide valley with high mountains in the distance (outlook); and (D) Forest glade, representing a clearing in the forest covered with spruce needles, low vegetation such as moss and wild blueberry bushes and surrounded by tall spruce trees (forest glade). In addition, the appearance of the start/end point of the forest path is shown (Figure 1e) and the location of the places (geographic coordinates 47.983662, 14.221897; Figure 1f). The connecting path covered a distance of about 4 km (3915 m) and a difference in altitude of 125 m (from 420 to 545 m). 

### 2.2. Data Collection and Study Sample

We collected data from June to October 2018. The researchers instructed and trained the local guides for the forest tour, following a standardized procedure. Participants took part in a guided forest tour with a minimum of two and a maximum of seven participants per group, lasting for about 2.5 h (M = 156.64 min, SD = 20.16). Each participant received a bottle of water prior to the tour. Sweets and the opportunity to take part in a raffle in fall 2018 were offered as compensation for participation. Measures took place at each of the four places (pre-visit) and also post-visit. At each place, the guides instructed participants to explore the place with all senses for 10 min. Participants were encouraged to walk around at site, to turn around, find a comfortable place to sit or lie down. They were asked not to talk to each other while experiencing the places. The guides rigorously checked the time and asked the participants to start their ratings after 10 min. The groups rated the selected places in a rotated order: 34 participants visited the places in order ABCD, 22 in order BCDA, and 34 in order DCBA. Complete rotation was not possible due to terrain-specific access restrictions. 

Participants were recruited through newspaper advertisements, the homepages of cooperating partners, and local clubs and unions. Inclusion criteria were age older than 16 years, ability to walk a forest trail for at least 2.5 h, immunization against tick-born encephalitis, and giving written informed consent prior to study participation. In total, 101 volunteers participated in the study; one dataset of a respondent younger than 16 years was excluded from analysis; one person took part in the tour, but did not want to participate in the study. The resulting sample was n = 99 (64% females, age range 16 to 81 years, mean age 43.15 years, SD 17.11). A total of 23% of the participants were visiting the Hallerwald for the first time, 50% visited it occasionally, 19% visited it more often than once a year, and 7% visited the forest more often than once a month. One person (1%) visited the forest several times a week.

We detected no pattern in missing values (<1.6%) using Little’s Missing Completely At Random (MCAR) Test (χ^2^ (398) = 385.60, p = 0.663) [28]. However, two clusters of missing values were detected visually. Due to two faulty copies of the questionnaire, the Restoration Outcome Scale (ROS) and Inclusion of Nature in the Self scale (INS) were missing for two participants. Ratings of the qualities of the path were missing for a further three participants; in one case, the page of the questionnaire was missing, in the other two cases, the study subjects did not fill out the respective page.

### 2.3. Measures

The questionnaire collected data on socio-demographic characteristics (i.e., age, sex, and frequency of visiting the Hallerwald) before entering the forest. Local guides used a smartphone app to count the steps during the tour. 

Perceived restorativeness potential (PRP) of the selected places in the forest was assessed using a short version of the Perceived Restorativeness Scale (PRS) comprising of eight items: four items each assessing being away and fascination [29,30]. Participants responded on an 11-point response format ranging from 0 “not true at all” to 10 “completely true”. Internal consistency (McDonald’s ω) was 0.88 [CI: 0.82; 0.91] for place A (mossy stones); 0.89 [CI: 0.85; 0.94], for place B (fern glade); 0.90 [CI: 0.87; 0.93] for place C (outlook), and 0.89 [CI: 0.83; 0.93] for place D (forest glade).

Vitality was assessed by three items of the vitality subscale of the German version of the SF-36 using a 11-point response format ranging from 0 “not true at all” to 10 “completely true” [31]. McDonald’s ω was 0.89 [CI: 0.82; 0.92] for place A; 0.94 [CI: 0.91; 0.96] for place B; 0.90 [CI: 0.87; 0.93] for place C; and 0.94 [CI: 0.91; 0.96] for place D. 

We assessed the potential for widening one’s mind by applying the newly developed WOM scale comprising of six items. The items described the potential of a place to foster reflective thinking, change of perspective, and generating new ideas. Participants responded on an 11-point answering format from 0 “not true at all” to 10 “completely true”. For statistical details, please see the Section 3.3. The authors will provide the WOM scale in German on request. 

We used a short version of the Positive Negative Affect Schedule (PANAS) to measure affect [32]. McDonald’s ω was 0.86 [CI: 0.76; 0.91] for the positive affect pre-visit and 0.84 [CI: 0.82; 0.93] post-visit. For negative affect pre-visit, ω was 0.84 [CI: 0.73; 0.91] and was 0.91 [CI: 0.71; 0.98] for the post-visit. Perceived stress was assessed using a single item asking participants how stressed they felt at the moment. Participants responded on an 11-point response format from 0 “not true at all” to 10 “completely true”. 

Perceived restoration was assed using three items of the ROS [33]. McDonald’s ω was 0.88 [CI: 0.83; 0.91]. 

Connectedness to nature (CN) was assessed using the INS [34]. Respondents marked one of seven graphics that best represented their relationship to nature. Connectedness with the Hallerwald, i.e., connectedness with the forest (CF) was assessed using a modified version of the INS by replacing the term “Nature” by “Hallerwald”.

Mindfulness was assessed using an 8-item version of the Freiburg Mindfulness Inventory [35]. Internal consistency (ω) was 0.76 [CI: 0.65; 0.83] pre- and 0.78 [CI: 0.70; 0.84] post-visit. 

After completing the tour, participants assessed the path (six items), and whether the participant would visit the forest again (1 item) or recommend the visit to friends (1 item). Participants answered on an 11-point scale from 0 “don’t agree” to 10 “completely agree”. 

Six lay persons pre-tested the questionnaire to ensure comprehensibility, face validity, and content validity of the survey and integrated their feedback in the final questionnaire.

### 2.4. Citizen Science Approach

Citizen science refers to the collaboration between members of the general public and professional scientists [6]. In this study, citizens and scientists collaborated in defining research questions, gathering information, collecting data, discussing results, drawing conclusions, and disseminating findings to the local community and beyond. In this study, the newly built research consortium agreed on a concept promoting the concept of “co-create to re-create” (i.e., remodeling the existing forest to maximize its recreational capacity).

### 2.5. Statistical Analysis

We evaluated the internal consistency using McDonald’s ω_h_ and provided a 95% confidence intervals (CI) [36]. For statistical analysis, we employed IBM SPSS Statistics Version 25.0 (IBM Corp., Armonk, USA) and the MBESS package for R statistics Version 3.4.1 (R Foundation for Statistical Computing, Vienna, Austria) [37,38]. To calculate the effect sizes (ES) for the dependent measures, an Excel-spreadsheet was used [39].

## 3. Results

### 3.1. Qualities of Places and the Connecting Forest Path

Table 1 shows the participants’ ratings of PRP, vitality, and WOM for the selected places and the forest as well as the respective post hoc comparisons with Bonferroni correction. We analyzed the qualities of selected places in the forest using RM-ANOVAs. As the assumption of homogeneity of variances was violated, we report Greenhouse–Geisser corrected tests for PRP (χ^2^(5) = 22.863, p < 0.001, ε = 0.88) and WOM (χ^2^(5) = 42.79, p < 0.001, ε = 0.76). Places differed statistically significantly in PRP (F(2.64, 258.99) = 28.722, p < 0.001) and WOM (F(2.28, 223.87) = 20.98, p < 0.001). No difference was found for vitality (F(2.78, 258.12) = 2.07, p = 0.109). All places received high scores in PRP. Fern glade and mossy stones scored higher than forest glade and outlook. Places did not differ in vitality. Fern glade showed the highest rating in WOM compared to the other places. 

Participants walked nearly two hours to reach all investigated places in the forest. The number of steps counted during the tour ranged from 5230 to 8411 (M = 6406.20, SD = 837.75). The path connecting these places was rated as not physically exhausting (M = 9.25, SD = 2.47), not too steep (M = 8.78, SD = 3.11), comfortable (M = 9.09, SD = 2.48), and safe (M = 8.49, SD = 2.42). The participants reported that walking in groups was experienced as pleasant (M = 9.48, SD = 0.95). People mostly agreed that they would visit the Hallerwald again (M = 8.70, SD = 1.94). Furthermore, they also agreed to recommend visiting the forest to friends (M = 8.62, SD = 1.82).

### 3.2. Changes in Participants’ Feelings and Perceptions

We analyzed the changes in positive and negative affect, perceived restoration, connectedness with nature and the forest, and mindfulness pre- and post-visit (Table 2). To increase readability and comparability of scores with different response categories, we reported all scores using the percentage of maximum possible (POMP) [40].

Paired sample t-tests revealed a statistically significant increase in the levels of positive affect, restoration connectedness with nature, connectedness with the forest, and mindfulness (all p < 0.001). Negative affect and perceived stress decreased. The most pronounced changes were the decrease in stress, the increase in perceived restoration, and the increase in CF.

Data analysis revealed an unexpected small increase in CF. Thus, we further analyzed whether familiarity with the Hallerwald might act as an intervening variable in this respect. Partial correlation analysis between the change in CF score and frequency of visits (r = 0.279, p < 0.001) was r = −0.001 (p < 0.996) when the pre-visit CF score was accounted for. Participants that were more familiar with the forest showed a higher initial value of CF and a lower change in CF. 

### 3.3. Psychometric Properties of the Widen One’s Mind (WOM) Scale

Principal axis factor analysis using varimax rotation assessed the factor structure of the six items forming the WOM scale (Table 3). Our data were suitable for factor analysis as the Kaiser–Meyer–Olkin measure of sampling adequacy score of >0.8 indicated an acceptable sample size, and the significant Bartlett’s test of sphericity (i.e., p < 0.001), indicated adequate inter-correlations between the items. The factor analysis revealed one factor explaining 70.79% of the variance. McDonald’s ω was 0.88 [CI: 0.82; 0.92] for place A; 0.90 [CI: 0.85; 0.93] for place B; 0.95 [CI: 0.93; 0.96] for place C; and 0.90 [CI: 0.86; 0.93] for place D.

## 4. Discussion

We conducted this case study to assess the potential for recreation, vitality, and to widen one’s mind at four places in the Hallerwald forest and the forest itself. In addition, we investigated changes in mood states, perceived stress, and restoration as well as connectedness and mindfulness before and after visiting the forest. Furthermore, we developed the WOM scale, which showed good scale characteristics. Participants of this cross-sectional study were a convenience sample of forest visitors that were recruited and guided through the Hallerwald by members of the working group of the local community of Adlwang. 

Regarding restorative potential of the places, all places received higher scores on the PRP compared to urban settings [41] and comparable scores obtained in a national park [5]. Notably, these studies used different versions of the PRS and differed in terms of sample characteristics, thus limiting the comparability of the results. The fern glade received the highest scores in PRP; however, it did not differ from the mossy stones. Thus, our assumption on the higher restorative potential of green places with water elements (“green and blue”) reported in prior research [19] was not supported by the data. This result could be associated with the water-like appearance of the fern glade. Additionally, the visual impression of water was impaired as the flow rate of the creek was very low due to the hot summer in Austria in 2018. The fern glade and the mossy stones showed higher scores compared to the forest glade and the outlook. These findings are in line with Carrus et al., who reported that a higher biodiversity increased perceived restorativeness [32]. The forest glade represented a typical spruce monoculture, whereas the fern glade appeared as an open place in the forest with different plants species. As the outlook was bordered by a road, the lower value of PRP could be explained by perceived traffic and traffic-related noise. Research on traffic noise exposure suggests an undermining effect of noise on restoration, health, and well-being [42]. 

Places did not differ significantly in their potential to foster vitality. Vitalizing effects of outdoor settings and forests, in particular, emerged after only 15 min of exposure in an experimental study [21]. With respect to vitality, we can conclude that all places performed equally in showing positive effects, and further, that these effects can be expected even after a short stay at any of the investigated places.

The local working group was interested in finding places in the forest that could widen one’s mind, leading to the creation of the WOM scale in German, which showed good psychometric characteristics in this study. The potential to widen one´s mind is closely related to constructs such as creativity [43] and reflective self-attention [44]. In addition, contact with nature was shown to increase the subjective ratings of creativity [16]. Future research should investigate how the potential to widen one’s mind of a particular place in a green space is associated with creativity. This potential could be systematically used to support creative processes in nature-based interventions for stress reduction and improvement of well-being in the sense of Green Public Health and Green Care [8]. 

While the WOM scale showed high face-validity, its value for assessing people–environment interaction needs further research. Validation studies should investigate its association with already existing, neighboring, or underlying constructs such as reflective thinking, rumination, mind wandering, or creativity [24,25,43,44,45]. We suggest follow-up studies to assess the performance of the WOM scale in other recreational outdoor as well as indoor environments.

Mood enhancement has also been found to be linked to visits of natural environments [16,17], especially forests [20,46,47]. In this case study, we found an increase in positive mood and a decrease in negative mood. This mirrors findings of other studies on mood enhancement during forest walks [47]. In contrast to findings of a meta-analysis conducted by McMahan and Estes, we found medium-size changes in both positive and negative affect [48]. While the magnitude of the change is comparable to the average effect found on positive mood in earlier studies, the decrease in negative mood was more pronounced in this study. The referred studies in the meta-analysis reporting the effect of brief exposure to natural environments on positive and negative affect did not provide a sound explanation for the distinct change in negative affect, as found in our study. There was no similarity in sample composition (mean age, sex) or other study characteristics (exposure type, environmental type, affect measure). Presumably, the pronounced decrease in negative affect in this study is specifically related to the forest setting or duration of exposure. However, it may also be that further, yet unknown phenomena caused the reduction in negative affect observed in our study.

With regard to restorative outcome, the reduction of perceived stress though forest visits is well documented [12]. In this study, two parameters mirrored this aspect. Perceived stress showed the most significant change (decrease of 55%) of all measured parameters during the visit. Perceived restorative outcome showed the second largest change (increase of 25%). These findings are in line with prior research indicating the stress-reducing and thus restorative effects of forest visits [16].

Furthermore, we found higher a CN and CF after the forest visit. CN increased by 9%. This is in line with prior research showing that contact to nature increased CN [5,49]. CN is associated with time spent in nature [50], well-being [51,52], and pro-environmental behavior [53,54]. Accordingly, a rise of 9% in CN after a single stay in the forest seems to be promising for health promotion as well as environmental protection. 

CF increased by 21%. In prior research, we found a larger change in CF (i.e., 30%), potentially due to aspects concerning the study population and familiarity with the Hallerwald [5]. Participants of the prior study were students and professors (mean age 28 years) taking part in an excursion to an Austrian national park. In contrast, participants in the current study were mainly local people (mean age 43 years). While the academic group predominantly visited the forest for the first time (73%), only 23% of participants in the current study visited the Hallerwald for the first time. The difference in the rise in CF might be due to a higher familiarity of the study population with the forest. Controlling for the pre-visit CF score extricated the correlation between the frequency of visits and the change in CF. We therefore conclude that the smaller rise in CF in this study can be attributed to the higher basic value in CF.

In a previous study, we investigated changes in mindfulness before and after visiting a national park [6]. Our current results on changes in mindfulness (increase of 7%) reproduced prior research. Mindfulness training has a long history in stress coping and psychotherapy [44]. A systematic review of nature-based mindfulness highlighted the beneficial effects of both contact to nature and mindfulness training. It further identified forest-like environments as superior in promoting health and well-being compared to other natural settings [45]. A recently published study showed that mindfulness training can further enhance the beneficial effects of contact to nature on mood and CN [36]. In the future, mindfulness training, especially in forests, may profit from restorative places. In light of the results from this case study, we suggest that mindfulness training in the Hallerwald might be conducted at the fern glade or the mossy stones due to their restorative potential. 

The study participants were a quite heterogeneous sample of healthy people from the general public of various ages, some were familiar with the forest and some were not. Familiarity turned out to be an intervening factor, probably not examinable in the homogenous samples as investigated in many studies on the beneficial effects of forests. Homogenous samples have been criticized due to limited generalizability [10]. Thus, our study might mirror the actual socio-demographic characteristics of forest visitors more when compared to other studies, thus showing a higher external validity. 

To our knowledge, this is the first study using a citizen science approach in a case study investigating the beneficial effects of forests. Inspired by ideas of the local working group in the sense of a co-creation process, we developed the WOM scale in German, which showed good psychometric characteristics in this research. However, further validation of the scale is needed in subsequent studies. Data collection took place during a guided forest tour in groups. Therefore, we were unable to separate the effects of guided tours from the effects of the forest as such. Future research could investigate the effects of a non-guided forest experience by using the outcome variables of our study (comparable to Korpela et al.) [16]. 

As a limitation of this study, we used self-reports. Self-reported data should be accompanied by tests of general mental capacity or physiological parameters in future studies [46]. We focused on connectedness with nature and the forest. Research on connectedness with nature showed correlations with health and well-being [40] and pro-environmental behavior [43]. However, we did not investigate the relationship with pro-environmental attitude or reported environmentally friendly behavior in this study. In future studies, this aspect should be addressed to fully investigate the broad range of the benefits of forest visits.

Selected places in the forest differed with respect to their affordances: Some were more relaxing than others; all supported vitality; the outlook allowed for a great view into the landscape. A forest visit of approximately 2.5 h resulted in improved mood, reduced stress, and perceived restoration. Additionally, connectedness with nature and the Hallerwald was increased. Prior research [39,41] already showed positive correlations of CN with well-being, sense of purpose, and time spent in nature, which are additional indicators for the promotion of health and of the quality of life. The findings of our study respond to various public health aims (e.g., increasing physical activity, reducing stress, and combating negative emotions). Finding relaxation, sense of purpose in life, and having a healthy lifestyle are further goals. Our study showed that visiting the forest in a group was suitable to contribute to all of these goals. Strengthened connectedness with the forest carries some additional important positive aspects for regional development, referred to as place attachment. Place attachment is correlated with a sense of personal identity and taking care of the place, also triggering the intention to revisit the place and recommend visiting it to a friend [55,56].

The aims of Green Care are to promote both the health and well-being of clients and diversification in agriculture and forestry [8]. As discussed above, health and well-being were promoted through the forest visit. Using forests not only for wood production, but also for the recreational purposes of forest visitors correspond to the idea of diversification. Physical activity in a recreational forest offering pleasant places describes a classic nature-based health intervention. Scientific evidence on the quality of the places and the forest as such builds the baseline for further interventions in the sense of Green Care and Green Public Health. These could include equipping places with relaxing features like wood benches or deckchairs. Aside from subjective measures, objective characteristics of the forest such as forest stand data should be taken into consideration [57]. Another option could be offering pedagogical, preventive, or therapeutic services that take up and amplify the inherent recreational qualities of the places. The availability of an evidence-based recreational forest may motivate health professionals to encourage patients to visit nature, as requested by Van den Berg [58].

The local working group intended to use the forest for the promotion of health and well-being to fulfill regional development plans. Before the planning interventions, we gathered and analyzed the baseline data. Hörnsten and Fredman described a recreational forest as a forest useful for spare time activities such as walking, skiing, or picking berries and mushrooms [23]. Our study explored a number of additional non-timber values of a forest including the potential for restoration and vitality increase and to widen one’s mind. For the visitors, the recreational forest provides beneficial mental effects such as deepening the connection with nature and the forest, positive affect, stress coping, and mindfulness in addition to physical activities. 

## 5. Conclusions

Our present study extends the understanding of the potential of visits to forests in increasing health and well-being and introduced the assets of citizen science. To facilitate regional development goals, we recommend evaluating places in forests regarding their potential effects on health and well-being as well as citizen participation before initiating extensive remodeling of the forest. We suggest preferably only using designated forests for nature-based health interventions. From a public health perspective, the creation of recreational forests could serve as health promoting interventions for local communities as well as tourists.

## Figures and Tables

**Figure 1 ijerph-17-01676-f001:**
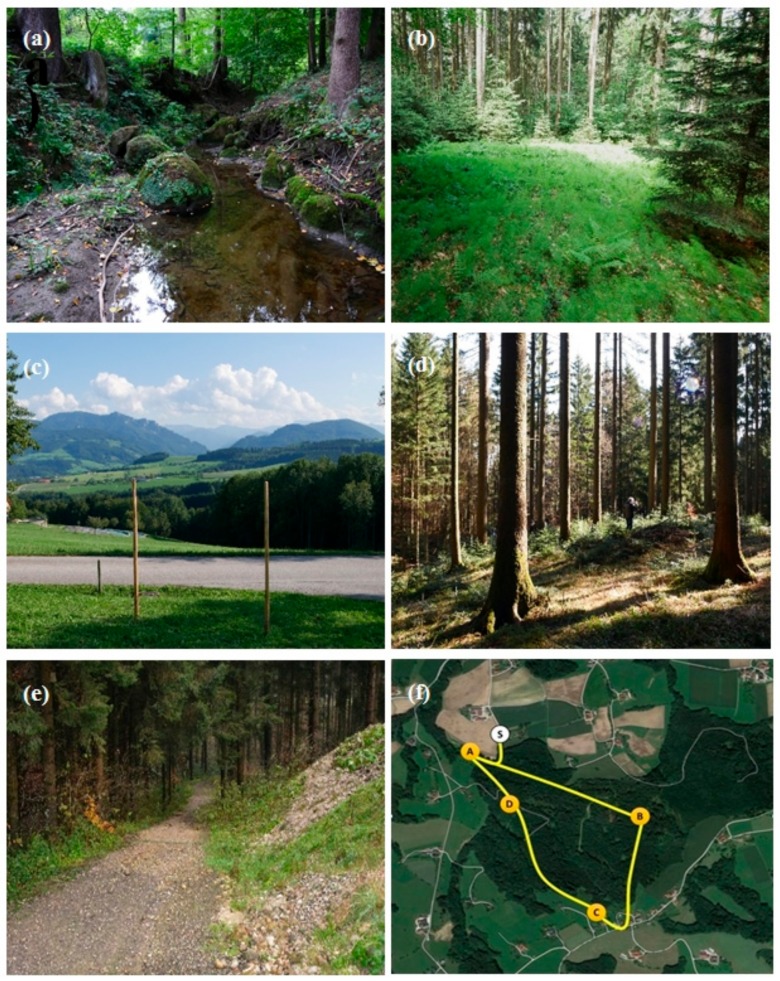
Impressions of the selected places in the forest: (**a**) Mossy stones (A in the map); (**b**) Fern glade (B); (**c**) Outlook (C); (**d**) Forest glade (D); (**e**) Start/end of the forest path (S); (**f**) Map of the places.

**Table 1 ijerph-17-01676-t001:** Qualities of the places and the forest.

Scores	(A) Mossy Stones		(B) Fern Glade		(C) Outlook		(D) Forest Glade		(A–D) Total Forest	
	Mean	SD	Mean	SD	Mean	SD	Mean	SD	Mean	SD
Perceived restorativeness potential	7.34 ^C,D^	1.42	7.70 ^C,D^	1.41	6.15 ^A,B^	1.69	6.65 ^A,B^	1.74	6.96	1.11
Vitality	6.08	1.99	5.79	2.15	5.90	2.17	6.29	2.08	6.03	1.66
Widen One’s Mind	7.67 ^B,C^	1.40	8.05 ^A,C,D^	1.38	6.69 ^A,B,D^	1.94	7.54 ^B,C^	1.55	7.49	1.16

Note: SD: standard deviation; ^A, B, C, D^ significant post-hoc comparisons (p < 0.05, Bonferroni corrected) for the places (A) Mossy stones, (B) Fern glade, (C) Outlook, (D) Forest glade.

**Table 2 ijerph-17-01676-t002:** Participants’ feelings and perceptions pre- and post-visit.

Scores	Pre-visit		Post-visit		p ^1^	Effect Size	Change (%)
	Mean	SD	Mean	SD			
Positive affect	59.01	18.83	69.40	17.02	<0.001	0.58 ^a^	−10.39
Negative affect	14.11	18.15	5.23	10.46	<0.001	0.59 ^b^	8.88
Perceived stress	75.92	26.41	21.16	12.52	<0.001	2.52 ^b^	54.76
Perceived restoration	61.05	20.32	85.74	10.66	<0.001	1.5 ^a^	−24.68
Connectedness with nature	59.61	23.55	68.77	22.05	<0.001	0.4 ^a^	−9.16
Connectedness with forest	29.51	26.71	50.53	27.45	<0.001	0.77 ^a^	−21.02
Mindfulness	55.36	15.03	61.89	14.87	<0.001	0.44 ^a^	−6.53

Notes: SD: standard deviation; ^1^ p values from t-tests; ^a^ Hedge’s g av (a) and ^b^ Hedge’s g rm (b); POMP-transformed scores.

**Table 3 ijerph-17-01676-t003:** Factor analysis of the Widen One’s Mind (WOM) scale at the places and the forest.

	A) Mossy Stones		B) Fern Glade		C) Outlook		D) Forest Glade		A-D) Total Forest	
**1. Items**	**fl**	**h^2^**	**fl**	**h^2^**	**fl**	**h^2^**	**fl**	**h^2^**	**fl**	**h^2^**
At this place one can										
let thoughts wander.	0.49	0.24	0.70	0.49	0.83	0.69	0.78	0.60	0.70	0.49
focus on the essentials.	0.86	0.75	0.84	0.71	0.79	0.62	0.86	0.73	0.90	0.80
gain a new perspective.	0.80	0.63	0.84	0.71	0.87	0.77	0.88	0.78	0.92	0.84
come to terms with oneself.	0.83	0.69	0.82	0.67	0.88	0.78	0.81	0.66	0.89	0.79
rearrange thoughts.	0.81	0.65	0.73	0.53	0.92	0.85	0.79	0.62	0.75	0.56
come up with new ideas.	0.65	0.42	0.76	0.58	0.84	0.71	0.61	0.38	0.87	0.76
**2. Measures**										
KMO	0.858		0.858		0.915		0.857		0.881	
Eigenvalue	3.375		3.692		4.420		3.775		4.248	
% of variance explained	65.24		61.53		73.70		62.92		70.79	

Note: fl: factor loadings; h^2^: communalities; KMO: Kaiser–Mayer–Olkin Measure of sampling adequacy.

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
