# Peer review of "Investigating the Qualities of a Recreational Forest: Findings from the Cross-Sectional Hallerwald Case Study"

_ijerph, 2020, doi:10.3390/ijerph17051676_

Round 1

Reviewer 1 Report

The paper is interesting as an idea but it requires a few small changes and improvements.

It would be better to revise the Title and make it more meaningful. Figures and Tables need to be properly formatted according to the format of the journal. i.e, In figure 1 what is D). The 2nd main limitation of the paper is that it lacks critical related work. The historical perspective should be discussed. The proposed study is not critically evaluated and compared to the related work/state-of-the-art and is not identified and discussed its drawbacks and limitations. Thus, it is not easy to assess the real contribution of the paper in the field and how much is efficient in the proposed study compared to related works. A clear assessment of the contribution of the authors when compared to existing approaches should be given. At the end of the literature review, you should have a summary of what the cited literature implies. This summary can be a paragraph to bring together all the pieces of literature you reviewed and to clarify where your research stands. The paper needs serious editing in terms of scientific writing style and some parts of the paper are misleading or have no logical order or flow. The paper must go through a rigorous editing and proofreading process before resubmission and strictly follow the Guideline for the Author(s).

Author Response

Comments to the Author

Reviewer #1:

Comments and Suggestions for Authors

The paper is interesting as an idea but it requires a few small changes and improvements.

It would be better to revise the Title and make it more meaningful. Figures and Tables need to be properly formatted according to the format of the journal. i.e, In figure 1 what is D). The 2nd main limitation of the paper is that it lacks critical related work. The historical perspective should be discussed. The proposed study is not critically evaluated and compared to the related work/state-of-the-art and is not identified and discussed its drawbacks and limitations. Thus, it is not easy to assess the real contribution of the paper in the field and how much is efficient in the proposed study compared to related works. A clear assessment of the contribution of the authors when compared to existing approaches should be given. At the end of the literature review, you should have a summary of what the cited literature implies. This summary can be a paragraph to bring together all the pieces of literature you reviewed and to clarify where your research stands. The paper needs serious editing in terms of scientific writing style and some parts of the paper are misleading or have no logical order or flow. The paper must go through a rigorous editing and proofreading process before resubmission and strictly follow the Guideline for the Author(s).

Authors‘ response:

We thank Reviewer #1 for the overall favorable evaluation of our manuscript!

+ We now present a modified title:

“Investigating the qualities of a recreational forest – Findings from the cross-sectional Hallerwald Case Study”

+ In response to the request, we removed the formatting errors in the table and the figure. We are truly sorry about that!

+ In response to the request concerning how the study contributes to the field, we added relevant literature.

* Bielinis, E.; Takayama, N.; Boiko, S.; Omelan, A.; Bielinis, L. The effect of winter forest bathing on psychological relaxation of young Polish adults. Urban For. Urban Green. 2018, 29, 276–283.

* Sonntag-Öström, E.; Nordin, M.; Lundell, Y.; Dolling, A.; Wiklund, U.; Karlsson, M.; Carlberg, B.; Slunga Järvholm, L. Restorative effects of visits to urban and forest environments in patients with exhaustion disorder. Urban For. Urban Green. 2014, 13, 344–354.

* Simkin, J.; Ojala, A.; Tyrväinen, L. Restorative effects of mature and young commercial forests, pristine old-growth forest and urban recreation forest - A field experiment. Urban For. Urban Green. 2020, 48, 126567.

* Ideno, Y.; Hayashi, K.; Abe, Y.; Ueda, K.; Iso, H.; Noda, M.; Lee, J.S.; Suzuki, S. Blood pressure-lowering effect of Shinrin-yoku (Forest bathing): A systematic review and meta-analysis. BMC Complement. Altern. Med. 2017, 17, 1–12.

* Nordström, E.M.; Dolling, A.; Skärbäck, E.; Stoltz, J.; Grahn, P.; Lundell, Y. Forests for wood production and stress recovery: trade-offs in long-term forest management planning. Eur. J. For. Res. 2015, 134, 755–767.

We also added the following paragraphe to the discussion section:

“The findings of this study suggest that in recreational forests, visitors experience beneficial mental effects such as stress reduction in addition to physical exercise. To facilitate regional development goals, we recommend evaluating places in the forests regarding potential effects on health and well-being as well as citizen participation before initiating extensive remodeling. Besides subjective measures objective characteristics of the forest, such as forest stand data should be taken into consideration”.

+ Also, we had the manuscript checked by an English native speaker. However, we are thankful for any additional hints to explain certain expressions that remain unclear to the reader.

Reviewer 2 Report

This is interesting topic but the manuscript needs to be improved to publish. My comments are below. 

Line 24-25: Please delete the sentence and write a conclusion of this study.

Introduction: Please cite more previous studies related to this study. 

Line 73-77: Hypothesis is addressed. Please rewriting and clarify the objectives of this study.

Line 187-188: move the sentences in the statistical analysis section.

Discussion: Please address study limitations and future study direction.

Author Response

Reviewer #2:

Comments and Suggestions for Authors

This is interesting topic but the manuscript needs to be improved to publish. My comments are below.

Line 24-25: Please delete the sentence and write a conclusion of this study.

Introduction: Please cite more previous studies related to this study.

Line 73-77: Hypothesis is addressed. Please rewriting and clarify the objectives of this study.

Line 187-188: move the sentences in the statistical analysis section.

Discussion: Please address study limitations and future study direction.

Authors‘ response:

We thank Reviewer #2 for the overall favorable evaluation of our manuscript!

+ We adapted the abstract accordingly:

“The findings of this study suggest that in recreational forests, visitors experience beneficial mental effects such as stress reduction in addition to physical exercise. To facilitate regional development goals, we recommend evaluating places in the forests regarding potential effects on health and well-being as well as citizen participation before initiating extensive remodeling. Besides subjective measures objective characteristics of the forest, such as forest stand data should be taken into consideration.”

+ In response to this request, we added several recently published studies.

* Bielinis, E.; Takayama, N.; Boiko, S.; Omelan, A.; Bielinis, L. The effect of winter forest bathing on psychological relaxation of young Polish adults. Urban For. Urban Green. 2018, 29, 276–283.

* Sonntag-Öström, E.; Nordin, M.; Lundell, Y.; Dolling, A.; Wiklund, U.; Karlsson, M.; Carlberg, B.; Slunga Järvholm, L. Restorative effects of visits to urban and forest environments in patients with exhaustion disorder. Urban For. Urban Green. 2014, 13, 344–354.

* Simkin, J.; Ojala, A.; Tyrväinen, L. Restorative effects of mature and young commercial forests, pristine old-growth forest and urban recreation forest - A field experiment. Urban For. Urban Green. 2020, 48, 126567.

* Ideno, Y.; Hayashi, K.; Abe, Y.; Ueda, K.; Iso, H.; Noda, M.; Lee, J.S.; Suzuki, S. Blood pressure-lowering effect of Shinrin-yoku (Forest bathing): A systematic review and meta-analysis. BMC Complement. Altern. Med. 2017, 17, 1–12.

* Nordström, E.M.; Dolling, A.; Skärbäck, E.; Stoltz, J.; Grahn, P.; Lundell, Y. Forests for wood production and stress recovery: trade-offs in long-term forest management planning. Eur. J. For. Res. 2015, 134, 755–767.

+ We reformulated the last paragraph of the introduction:

“The current field study aimed at I) assessing the qualities of typical places in the forest for their potential to increase recreation and vitality and to widen one's mind and II) investigating changes in mood-states, perceived stress, restoration, connectedness, and mindfulness before and after visiting the forest (pre post). Based on prior research we hypothesized that all places may reach high scores in their potential to increase restoration and vitality and to widen one´s mind, and further, that these scores differ regarding the qualities of the various scenic characteristics [5]. Also, we hypothesized that positive affect, restoration, connectedness with nature and the forest, and mindfulness may increase during visiting the forest. In contrast, we expected negative affect and perceived stress to decrease in participants. For measuring the potential for widen one's mind, we III) developed the so-called Widen One’s Mind (WOM) scale and tested its psychometric properties.”

+ As requested, we moved the phrase to the statistics section.

“We analyzed the qualities of selected places in the forest using repeated measures analysis of variance (rmAnova).”

+ Without a designate section on “study limitations”, we discuss those and also bring according suggestions for further studies.

Reviewer 3 Report

This study has merits and can be a useful contribution to the field, if the following items are addressed:

Use of undefined terms in abstract (“the connecting path”, “Green Care”). Abstract should be written as stand-alone source of information, all terminology should be able to be understood be readers, independent of reading full manuscript. Overall review of English language for grammar and typographical errors. Some cases pointed out in PDF comments.  There is question if the term “Green Public Health” is a defined term as referenced in Sempik et al 2010 (Reference #8). This is not clear from the reference title alone.  Please clarify to the reviewer that “Green Public Health” is a term defined explicitly in this citation. The concept of “Widen One’s Mind” is not defined anywhere. This would be problematic in any study.  This is especially true in a study that aims to establish a new instrument for measuring this subjective psychological construct.  Much more needs to be written about this concept, both in the Introduction and Section 2.3, about what exactly is being measured and what is the purpose of measuring it for this to be an appropriate effort put forward in this manuscript.  In Section 2.1, it is not clear how the four places were selected for inclusion. If there was a formal (or informal) process of selecting these sites (versus other possibly considered sites, for example), this should be described in greater detail. Approximately how much land area does each of the four sites occupy? Inclusion of a map of the area would be helpful, to provide location of the four sites relative to each other, and the relative distances between them (and each other). Are they meters apart?  Kilometers?  Walking distance has a significant impact on all outcome measures used in this study. In addition, please state how much distance was covered walking during the visit to these four sites. In Discussion, it should be stated that reference of comparative PRP scores of urban and national parks are not within-subjects for this study, but are different groups of participants from different studies. Thus results may not be generalizable ie it is not the same people taking PRP at all locations.  Concept of “sustainability” is mentioned without any discussion of how the purpose or outcomes of this study relate to this concept. Please clarify what this study has to do with sustainability. It is inappropriate to make the recommendation of re-evaluation of forests for non-timber measures based on the findings of the current study. Much research in this field has been conducted, and in this area the current study is at best an average-level addition. If changes to forest assessment policy are recommended, it would be best to base that on a total body of work in a field, and not one’s personal study of 99 people.  Reference to “power places” is unknown to most researchers and readers in this field. Include citation(s). Inclusion of the concept of “economic boost” in the closing sentence of the paper is out of place, when this concept has not been discussed anywhere else in the paper, certainly not as one of the primary aims. Recommend developing this concept more or omitting it from the concluding sentence. 

See attached PDF for specific commentaries.

Author Response

Reviewer #3:

Comments and Suggestions for Authors

This study has merits and can be a useful contribution to the field, if the following items are addressed:

Use of undefined terms in abstract (“the connecting path”, “Green Care”). Abstract should be written as stand-alone source of information, all terminology should be able to be understood be readers, independent of reading full manuscript.

Authors‘ response:

We thank Reviewer #3 for the overall favorable evaluation of our manuscript!

We adapted the abstract accordingly:

“Prior research shows that forest contribute to human health and well-being. In this sense, this cross-sectional case study adopting principles of citizen science aimed at assessing the restorative potential of places in the Hallerwald, an Austrian community forest. A convenience sample of adult forest visitors (n=99, 64% females) completed a survey during a guided 2.5 hour forest tour. The German questionnaire assessed qualities of defined places in the forest. Also, we investigated changes in mood states, perceived stress, restoration, connectedness, and mindfulness before and after visiting the forest. In cooperation with a local working group, we developed a new Widen One´s Mind (WOM) scale, which showed good scale characteristics. All places received high scores in their potential to increase restoration, vitality, and to widen one´s mind. Positive affect, restoration, connectedness with nature and the forest, and mindfulness increased pre-post visits, whereas negative affect and perceived stress decreased. The findings of this study suggest that in recreational forests, visitors experience beneficial mental effects such as stress reduction in addition to physical exercise. To facilitate regional development goals, we recommend evaluating places in the forests regarding potential effects on health and well-being as well as citizen participation before initiating extensive remodeling.”

Reviewer´s comment:

Overall review of English language for grammar and typographical errors. Some cases pointed out in PDF comments.

Authors‘ response:

We agree and are thankful for the indicated errors. Also, we had the manuscript rigorously checked by an English native speaker. However, we are thankful for any additional hints to explain certain expressions that remain unclear to the reader.

Reviewer´s comment:

There is question if the term “Green Public Health” is a defined term as referenced in Sempik et al 2010 (Reference #8). This is not clear from the reference title alone.  Please clarify to the reviewer that “Green Public Health” is a term defined explicitly in this citation.

Authors‘ response:

The reference described the term Green Care. We now refer to Haluza et al. (2014) where “Green Public Health” is described as using all kinds of natural resources for public health promotion. These may encompass interventions on an individual or group level such as therapeutic horticulture and animal assisted interventions as well as on a community or population level.

Reviewer´s comment:

The concept of “Widen One’s Mind” is not defined anywhere. This would be problematic in any study.  This is especially true in a study that aims to establish a new instrument for measuring this subjective psychological construct.  Much more needs to be written about this concept, both in the Introduction and Section 2.3, about what exactly is being measured and what is the purpose of measuring it for this to be an appropriate effort put forward in this manuscript. 

Authors‘ response:

 We developed this new scale as established scales and constructs did not measure what the project partners, i.e. the local working group of the Hallerwald study, were focused on assessing characteristics of persons and not places and, in terms of face validity, project partners described something different than established scales measured (on item level). We adapted the following passages:

“While there is abundant scientific evidence on the beneficial effects of forests, literature on the potential of forests to widen a visitor’s mind is scarce. This concept referred to the idea of fostering contemplation, initiating change in perspective, and developing new perspectives and ideas from the forest visitor´s perception. Similar processes have been described in qualitative research on forest based rehabilitation: Patients’ mental processes changed from negative to positive thoughts when choosing a favorite place in the forest, finding peace of mind, and reflective thinking. Established instruments for the assessment of reflective thinking focused on assessing characteristics of persons and not places and were not able to sufficiently capture the notion of widen one’s mind. In a co-creation process, we thus developed a respective new scale with the project partners.” (intro)

“The items aimed at describing the potential of a place to foster reflective thinking, change of perspective, and generating new ideas.” (methods)

“While the WOM scale showed high face-validity, its value for assessing people-environment interaction needs further research. Validation studies should investigate its association with already existing, neighboring or underlying constructs such as reflective thinking, rumination, mind wandering, or creativity. (discussion)

Reviewer´s comment:

In Section 2.1, it is not clear how the four places were selected for inclusion. If there was a formal (or informal) process of selecting these sites (versus other possibly considered sites, for example), this should be described in greater detail. Approximately how much land area does each of the four sites occupy? Inclusion of a map of the area would be helpful, to provide location of the four sites relative to each other, and the relative distances between them (and each other). Are they meters apart?  Kilometers?  Walking distance has a significant impact on all outcome measures used in this study. In addition, please state how much distance was covered walking during the visit to these four sites.

Authors‘ response:

In response to this request, we added a map indicating the places, the degree of latitude and longitude (i.e. geographic coordinates 47.983662, 14.221897).

Methods:

 “The project partners selected four distinct places that represented different scene types of the Hallerwald and should foster restoration and vitality, and have the potential to widen one´s mind. Places should be easily accessible by foot and not hazardous e.g. due to high risk of stumbling.“

“Also, appearance of the start/end point of the forest path is shown (Fig. 1 e) and the location of the places (geographic coordinates 47.983662, 14.221897; Fig. 1 f). The connecting path was about 4 km (3915 m) long; however, the walking distance was approximately 4.5 km as some passages were walked back and forth. “

Reviewer´s comment:

In Discussion, it should be stated that reference of comparative PRP scores of urban and national parks are not within-subjects for this study, but are different groups of participants from different studies. Thus results may not be generalizable ie it is not the same people taking PRP at all locations. 

Authors‘ response:

We adapted the paragraph in the discussion:

“Notably, these studies used different versions of the PRS and differed in terms of sample characteristics, thus limiting comparability of results.”

Reviewer´s comment:

Concept of “sustainability” is mentioned without any discussion of how the purpose or outcomes of this study relate to this concept. Please clarify what this study has to do with sustainability. It is inappropriate to make the recommendation of re-evaluation of forests for non-timber measures based on the findings of the current study. Much research in this field has been conducted, and in this area the current study is at best an average-level addition. If changes to forest assessment policy are recommended, it would be best to base that on a total body of work in a field, and not one’s personal study of 99 people. 

Authors‘ response:

We agree and deleted the notion of sustainability from the manuscript. It might be that the concept of sustainability is not mirrored in the German language to this extent. As for non-timber measures, we deleted this phrase: “In view of the findings of this study, we recommend evaluating forests with respect to their effects before setting non-timber measures.”

Reviewer´s comment:

Reference to “power places” is unknown to most researchers and readers in this field. Include citation(s). Inclusion of the concept of “economic boost” in the closing sentence of the paper is out of place, when this concept has not been discussed anywhere else in the paper, certainly not as one of the primary aims. Recommend developing this concept more or omitting it from the concluding sentence. 

Authors‘ response:

We agree and include a suitable reference for power places.

In response to the comment regarding economic boost as well as power places, we agree and thus deleted these ideas.

“From a Public Health perspective, the creation of recreational forests could serve as health promoting intervention for local communities as well as tourists.”

See attached PDF for specific commentaries.

Reviewer´s comment:

Abstract

Authors‘ response:

We modified the abstract, see above.

Reviewer´s comment:

When did this occur?  In the 1980s when shinrinyoku originated?  Last year?  Sometime in-between?

Authors‘ response:

Forest bathing emerged in the last decade, with a steep increase in attention in the last few years. We now modified the phrase as follows:

“Originating from East-Asia, the concept of Shinrin-Yoku, which means taking in the atmosphere of the forest, reached Austria and received high attention in forestry and tourism in the late 2010s.”

Where indicated, we reworded the phrase:

e.g

*Targeting locals, clients from the nearby health resort, and tourists, the Hallerwald as an acknowledged recreational forest should provide recreation, vitality, and inspire visitors to widen their mind.

To

As an acknowledged recreational forest, the Hallerwald should provide recreation, vitality, and inspire visitors to widen their mind for the local community and tourists alike.

*Visiting restorative places in a forest increased connectedness with nature and the forest and mindfulness.

To

Restorative places in a forest increased connectedness with nature and the forest and mindfulness.

*fostering contemplation of visitors:

deleted

*aims of green care (discussion)

We added the citation of the framework of Green Care (Sempik et al. 2010).

Round 2

Reviewer 1 Report

The manuscript has been improved a lot as per the suggestions provided. However, the scientific writing style and English Language are still very poor and I strongly recommend getting the manuscript proofread again by professional/3rd party Language editors.

Author Response

Comments and Suggestions for Authors:

The manuscript has been improved a lot as per the suggestions provided. However, the scientific writing style and English Language are still very poor and I strongly recommend getting the manuscript proofread again by professional/3rd party Language editors.

Authors`reply:

We thank Reviewer 1 for the valuable evaluation of the revised version of our manuscript.

In response to this comment on English language and style we have sent the manuscript to a - now different - native speaker, a professor from the UK associated with our university, to check for typos and style. We are truly sorry for the typos!

We provide now a new version of the manuscript with changes highlighted in red color. However, we are happy for any additional concrete hints about style issues that we should correct before the manuscript is eventually spell-checked by the editorial team in the course of the pre-publication adaptations.

Reviewer 2 Report

Thank you for the revised manuscript. The manuscript was improved by following the comments.

Author Response

Comments and Suggestions for Authors:

Thank you for the revised manuscript. The manuscript was improved by following the comments.

Authors`reply:

We thank Reviewer 3 for the valuable evaluation of our manuscript.

Reviewer 3 Report

Thank you for your detailed attention and response to comments.  This paper is now ready for publication and will be a contribution to the field.

Author Response

Comments and Suggestions for Authors:

Thank you for your detailed attention and response to comments. This paper is now ready for publication and will be a contribution to the field.

Authors`reply:

We thank Reviewer 3 for the valuable evaluation of our manuscript.